# COUNTERBALANCING TEACHER: REGULARIZING BATCH NORMALIZED MODELS FOR ROBUSTNESS

## ABSTRACT

Batch normalization (BN) is a ubiquitous technique for training deep neural networks that accelerates their convergence to reach higher accuracy. However, we demonstrate that BN comes with a fundamental drawback: it incentivizes the model to rely on frequent low-variance features that are highly specific to the training (in-domain) data, and thus fails to generalize to out-of-domain examples. In this work, we investigate this phenomenon by first showing that removing BN layers across a wide range of architectures leads to lower out-of-domain and corruption errors at the cost of higher in-domain error. We then propose the Counterbalancing Teacher (CT) method, which leverages a frozen copy of the same model without BN as a *teacher* to enforce the student network's learning of robust representations by substantially adapting its weights through a consistency loss function. This regularization signal helps CT perform well in unforeseen data shifts, even without information from the target domain as in prior works. We theoretically show in an overparameterized linear regression setting why normalization leads a model's reliance on such in-domain features, and empirically demonstrate the efficacy of CT by outperforming several methods on standard robustness benchmark datasets such as CIFAR-10-C, CIFAR-100-C, and VLCS.

## 1 INTRODUCTION

Batch normalization (BN), a neural network layer that normalizes input features by aggregating batch statistics during training, is a key component for accelerating convergence in the modern deep learning toolbox (Ioffe and Szegedy, 2015; Santurkar et al., 2018; Bjorck et al., 2018). It plays a critical role in stabilizing training dynamics for large models optimized with stochastic gradient descent, and has since spurred a flurry of research in related modifications (Ba et al., 2016; Kingma and Ba, 2014) and its understanding (Gitman and Ginsburg, 2017; Santurkar et al., 2018; Luo et al., 2018; Kohler et al., 2018).

Despite its advantages, BN has recently been shown to be a source of vulnerability to adversarial perturbations (Galloway et al., 2019; Benz et al., 2021). In our work, we take this observation one step further and demonstrate that BN also compromises a model's out-of-domain (OOD) generalization capabilities. Specifically, we demonstrate that normalization incentivizes the model to exploit highly predictive, low-variance features (Geirhos et al., 2018; 2020), that lead to poor classification accuracy when the test environment differs from that of training. Given the widespread use and benefits of normalization, we desire a way to mitigate such drawbacks in models trained with BN.

To better understand this phenomenon, we investigate the effect of normalization in over-parametrized regimes, where there exist multiple solutions and inductive bias (e.g., minimizing the norm of the weights) significantly impacts the estimated parameters. Similar to recent work in the theory of deep learning (Khani and Liang, 2021; Raghunathan et al., 2020; Nakkiran, 2019; Hastie et al., 2019; Liang et al., 2020), we study the min-norm solution in over-parametrized linear regression. Without normalization, the inductive bias selects a model that fits training data and minimizes a fixed norm independent of data; with normalization, the same inductive bias selects a model that minimizes a data-dependent norm, leading the model to rely more on low-variance features. While such highly predictive features yield better performance in-domain where the features do not vary significantly, they cause performance to plummet in OOD settings (e.g. data corruptions or missing features). This

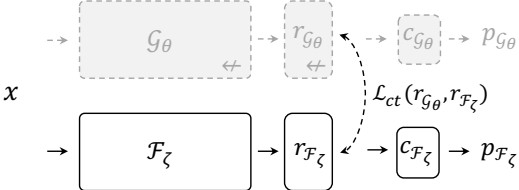

Figure 1: The Counterbalancing Teacher (CT) model's architecture. In the first step, the teacher encoder (i.e., unnormalized) $\mathcal{G}_\theta$ is trained to map input $x$ into a label. In the second step, we freeze $\mathcal{G}_\theta$, remove its classification head $c_{\mathcal{G}_\theta}$, and train the student encoder (i.e., with batch-norm) $\mathcal{F}_\zeta$ while regularizing its learned representations $r_{\mathcal{F}_\zeta}$ using distance function $\mathcal{L}_{ct}$ and $r_{\mathcal{G}_\theta}$. Stopping gradients operation is shown by $\nleftarrow$. Classifier heads and class probabilities are denoted by $c$ and $p$, respectively. During inference, we only use $\mathcal{F}_\zeta$.

is in direct contrast to models trained without BN that assign equal weight to *all* input features, which help to reduce their overfitting on the training set.

Drawing inspiration from our observation and the knowledge distillation literature (Hinton et al., 2015; Romero et al., 2014), we propose a simple teacher-student model to combine the best of both worlds: we leverage features derived both from a network without BN (teacher) and its clone with BN (student) to learn representations that achieve high standard and robust accuracies. We incorporate a regularization term in the loss function which encourages the features learned from the student encoder to have similar statistics and structure to those learned from the teacher; we name this model the Counterbalancing Teacher (CT) and show that it helps in achieving both higher robust and clean accuracy compared to a (batch) normalized model. In particular, CT retains good performance in OOD settings even *without* knowledge of statistics of the new domain.

Our results mark a significant improvement over prior works, which have tackled similar problems by either: (a) modifying the statistics of a trained model (Schneider et al., 2020; Benz et al., 2021) using privileged information from the target domain; or (b) augmenting the training data using a set of predefined corruption functions (Hendrycks et al., 2019b). As recent studies (Vasiljevic et al., 2016; Geirhos et al., 2018; Taghanaki et al., 2020) show that such approaches often fail to generalize due to the tendency of neural networks to memorize data-specific properties, this motivates a shift towards developing models that are inherently robust, independent of data augmentation or input transformation.

Empirically, we demonstrate that CT outperforms most existing data augmentation-based techniques and covariate shift adaption-based methods (which require information from the test set) on mean corruption error on CIFAR10-C and CIFAR100-C (Hendrycks and Dietterich, 2019), and achieves state-of-the-art performance in domain generalization on the VLCS dataset (Torralba and Efros, 2011). We further test CT on corrupted 3D point-cloud data (Taghanaki et al., 2020) and show it outperforms existing methods in terms of mean classification accuracy over multiple test sets. To the best of our knowledge, this is the first work to explore both theoretically and empirically why BN leads to a model's over-reliance on frequent, low-variance features, which can adversely affect its performance on the downstream classification task. This is also the first work to present a robust representation learning framework for common input distortions without additional data augmentation strategies or information derived from the target domain.

In summary, our contribution is threefold:

1. We provide theoretical justifications for why normalization encourages a model to exploit low-variance features, and empirically evaluate how this behavior can adversely affect downstream classification accuracy.

2. We propose CT, a representation learning approach demonstrating that regularizing representations of a batch-normalized network (i.e., student) using those from an unnormalized copy (i.e., teacher) can significantly improve a model's robustness.

3. We experimentally verify the robustness of the representations learned by CT to input distortions and domain shift on a variety of tasks and models.

## 2 PROBLEM STATEMENT AND ANALYSIS

We assume a supervised classification setting: given an input variable $x \in \mathcal{X} \subseteq \mathbb{R}^d$, and a set of corresponding labels $y \in \mathcal{Y} = \{1, ..., k\}$, we aim to learn a classifier $f_\zeta : \mathcal{X} \longrightarrow \mathcal{Y}$ by minimizing the empirical risk:

$$\zeta = \operatorname*{argmin}_\zeta \mathbb{E}_{x,y \sim p_d(x,y)}[\ell(x, y; \zeta)] \approx \operatorname*{argmin}_\zeta \sum_{i=1}^n \ell(f_\zeta(x_i), y_i) \tag{1}$$

Here $p_d(x, y)$ is the underlying joint distribution where the dataset $\mathcal{D} = \{(x_i, y_i)\}_{i=1}^n$ is sampled from. In the following sections, we first discuss the effect of (batch) normalization on the solutions found in underspecified regimes, then elaborate on our approach to optimize the aforementioned empirical risk. In this context, we refer to a problem as underspecified or overparameterized when degrees of freedom of a model is larger than the number of training samples.

### 2.1 THE EFFECT OF NORMALIZATION IN OVERPARAMETRIZED REGIMES

Modern deep learning frameworks usually incorporate many parameters (often larger than the number of training data points), which lead to underspecified regimes. In other words, many distinct solutions solve the problem equally i.e., have the same training or even held-out loss (D'Amour et al., 2020). In the underspecified regime, the inductive bias of the estimation procedure, such as choosing parameters with the minimum norm, significantly impacts the estimated parameters. In such regimes, we show that normalizing data incentivizes the model to rely on features with lower variance. We analyze the effect of normalization on the min-norm solution in overparametrized noiseless linear regression. This setup has been studied in many recent works for understanding some phenomena in deep networks (Khani and Liang, 2021; Raghunathan et al., 2020; Nakkiran, 2019).

Let $X \in \mathbb{R}^{n \times d}$ denote training examples and $Y \in \mathbb{R}^n$ denote their target. Considering that we are in an over parametrized regime ($d > n$), there should be an equivalence class of solutions. We assume that the inductive bias of the model is to choose the min-norm solution (the parameter with the minimum $\ell_2$ norm). This is in line with the recent speculation that the inductive bias in deep networks tends to find a solution with minimum norm (Gunasekar et al., 2018). One can show that the convergence point of gradient descent run on the least-squares loss is the min-norm solution.

Without normalization, the model chooses the min-norm solution which fits the training data:

$$\hat{\zeta} = \operatorname*{arg\,min}_\zeta \|\zeta\|_2^2 \quad s.t. \quad X\zeta = Y. \tag{2}$$

Now we observe how normalization changes the estimated parameters. Let $U$ be a diagonal matrix where $U_{ii}$ denotes the standard deviation of the $i^{\text{th}}$ feature. By normalization, we transform $X$ to $XU^{-1}$ (for simplicity, we assume the mean of each feature is 0, we can show that transforming points do not change the estimated parameter, see Appendix C.1 for details). In this case the model estimates $\hat{\beta}$ as follows:

$$\hat{\beta} = \operatorname*{arg\,min}_\beta \|\beta\|_2^2 \quad s.t. \quad XU^{-1}\beta = Y, \tag{3}$$

and since we normalize data points at the test time as well, the estimated parameter used for prediction at the test time is $\hat{\theta} = U^{-1}\hat{\beta}$. Substituting $\theta$ instead of $U^{-1}\beta$ (thus $U\theta = \beta$), we can write the equal formulation of 3 as:

$$\hat{\theta} = \operatorname*{arg\,min}_\theta \|U\theta\|_2^2 \quad s.t. \quad X\theta = Y. \tag{4}$$

For the same equivalence class of solutions a model with normalization (4) chooses different parameters in comparison to a model without normalization (2). In particular, 2 chooses an interpolant with a minimum *data independent* norm. On the other hand, 4, chooses an interpolant with a minimum *data-dependent* norm, which incentives the model to assign higher weights to low variance features. Note that projection of $\hat{\theta}$ and $\hat{\zeta}$ is the same in column space of training points. Formally if $\Pi = X^\top(XX^\top)^{-1}X$ denote the column space of training points then $\Pi\hat{\theta} = \Pi\hat{\zeta}$. However, their projections to the null space of training points $(I - \Pi)$ are different. As a result as we have

more data (smaller null space), $\hat{\theta}$ and $\hat{\zeta}$ become closer, and converge when $n > d$. Our analysis hold for classification with max-margin, we only need to substitute $X\theta = Y$ by $Y \odot X\theta \geq 1$ (see Appendix C.2 for details).

We conjecture that minimizing the data-dependent norm in each layer leads to reliance on low variance (frequent) features, which can result in a better in-domain generalization as these feature do not exhibit high variations. Nonetheless, in a new domain where some (or all) of the training-domain features are missing or altered (e.g., when there is some data-agnostic corruption such as Gaussian noise), 2 performs better as its inductive bias is data independent. How should we change the regularization such that it selects for a model that performs well both in- and out-of-domain? Inspired by this analysis, we introduce a simple, yet powerful two-step approach that combines the normalized and unnormalized copies of the same network for robust representation learning.

## 3 COUNTERBALANCING TEACHER (CT)

### 3.1 TRAINING THE TEACHER ENCODER (I.E. UNNORMALIZED ENCODER)

For an arbitrary neural network encoder $\mathcal{F}$ parametrized by $\zeta$ with one or more batch normalization layers, we first create a clone of the network. We refer to the cloned network as $\mathcal{G}$ and its parameters as $\theta$. We then remove all the batch norm layers from the cloned network and train it using the classification ($k$ class) objective of the task in hand $\mathcal{L}_{cls}^{\mathcal{G}}$. In our experiments, we minimize the cross-entropy loss as a supervised classification objective:

$$\mathcal{L}_{cls}^{\mathcal{G}}(p_{\mathcal{G}_\theta}, y) = -\sum_i^k y^i \log(p_{\mathcal{G}_\theta}^i), \tag{5}$$

henceforward we refer to $\mathcal{G}_\theta$ as the Counterbalancing Teacher or simply CT, and $\mathcal{F}_\zeta$ as the Student.

### 3.2 REGULARIZING FOR ROBUSTNESS USING THE COUNTERBALANCING TEACHER (CT) METHOD

We subsequently use CT with frozen parameters to regularize the training objective of the Student network. Inspired from the work of Huang and Belongie (2017), we propose the following objective to regularize the representations of $\mathcal{F}_\zeta$ (i.e. Student) using:

$$\mathcal{L}_{ct}\left(r_{\mathcal{G}_\theta}, r_{\mathcal{F}_\zeta}\right) = \frac{1}{h}\sum_{i=1}^h \left(r_{\mathcal{G}_\theta}^i - r_{\mathcal{F}_\zeta}^i\right)^2 \tag{6}$$

$\mathcal{F}_\zeta$ and $\mathcal{G}_\theta$ map the input $x$ to latent representations $r_{\mathcal{F}_\theta} \in \mathbb{R}^h$ and $r_{\mathcal{G}_\theta} \in \mathbb{R}^h$ respectively. $h$ denotes the dimension of feature vectors. Finally, we add $\mathcal{L}_{ct}$ to the cross entropy objective and minimize:

$$\mathcal{L}(r_{\mathcal{G}_\theta}, r_{\mathcal{F}_\zeta}, p_{\mathcal{F}_\zeta}, y) = \mathcal{L}_{cls}^{\mathcal{F}}(p_{\mathcal{F}_\zeta}, y) + \lambda\mathcal{L}_{ct}(r_{\mathcal{G}_\theta}, r_{\mathcal{F}_\zeta}) \tag{7}$$

To calculate the first and and second terms in Equation 6 (RHS), we $l_2$ normalize both $r_{\mathcal{G}_\theta}$ and $r_{\mathcal{F}_\zeta}$. Another variant of the model would be to train both $\mathcal{G}_\theta$ and $\mathcal{F}_\zeta$ simultaneously, however, as $\mathcal{F}_\zeta$ converges faster, it dominates $\mathcal{G}_\theta$. We found the current 2-steps setting working better i.e., training and freezing $\mathcal{G}_\theta$ in the first step. Similar to knowledge distillation, a limitation of our CT method can be training an extra encoder as a regularizer, however, note that the encoder is removed during inference, therefore the processing time at inference will be similar to that of the original model. The $\lambda$ in Eq. 7 controls the amount of regularization.

## 4 EXPERIMENTAL RESULTS

In this section, we are interested in empirically investigating the following questions:

    1. Do batch and other normalization techniques lead to learning frequent low variance features?

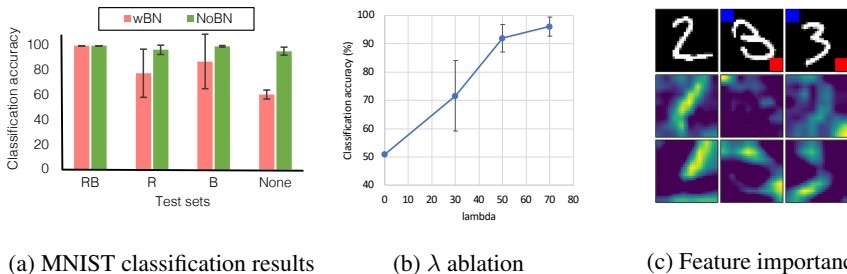

(a) MNIST classification results     (b) $\lambda$ ablation     (c) Feature importance

Figure 2: (a) In the MNIST classification task, the model without BN (NoBN) achieves high classification accuracy even if the frequent training features are missing during inference. (b) Ablation results on the "None" test set. (c) The model with BN ($2^{nd}$ row) puts more emphasis on the red/blue squares as they are the frequent features while the NoBN model ($3^{rd}$ row) does not.

2. Would a trained model with normalization fail if the dominant features are missing or corrupted, and how would an unnormalized network behave in the same scenario?

3. Does our CT method lead to a reasonable in- and out-of-domain accuracy as well as low corruption error across different architectures and data modalities?

## 4.1 BATCH NORMALIZATION IS A CAUSE OF FAILURE WHEN FREQUENT INPUT FEATURES ARE MISSING

We start with a synthetic experiment to test the hypothesis that batch normalization leads to learning low variance dominant features. We design a binary classification problem by selecting two digits of "2" and "3" from the MNIST dataset and add a red and a blue square to all samples of class "3" (Figure 2c first row) as "frequent" features.

We examine this hypothesis both qualitatively (visualizing the important input features for the trained classifiers using GradCAM (Selvaraju et al., 2017)) and quantitatively by evaluating on multiple test sets: 1) *RB*: similar to training samples, digit "3" includes both the red and blue squares, 2) *R*: digit "3" includes only red square, 3) *B*: digit "3" includes only blue square, and 4) *None*: contains no square.

Ideally, a classifier should not rely *only* on the dominant features to predict the class. However, as shown in the second row of Figure 2c, the network trained with batch normalization quickly picks up those low variance clues, which in this case is either the red or the blue square. However, the same network without normalization takes into account other features as well. We verify this by quantifying classification accuracy on the different test sets. As demonstrated in Figure 2a, since the model relies mostly on the colored squares, it drastically fails when those features are missing at test time (the "None" test set) which is not the case for the network without normalization.

## 4.2 BATCH NORMALIZATION IS A CAUSE OF FAILURE WHEN INPUT FEATURES ARE CORRUPTED

In this experiment, we focus on a scenario where models are trained with clean data while the test data is shifted by common data corruptions. Here, we use the WideResNet 40-2 and AllConvNet architectures and CIFAR-10-C dataset to be consistent with the work of Hendrycks et al. (2019b). For simplicity and to directly study the effect of normalization under data shifts we do not use dropout and data augmentations in this experiment. We further replace batch normalization with other common normalization techniques such as layer normalization (LN) (Ba et al., 2016) and instance normalization (IN) (Ulyanov et al., 2016). As shown in Figure 3, the model with no normalization (NN) outperforms all the three types of normalization when evaluated on corrupted data. Whereas, on the clean data all three techniques do generally well. With our CT model, corruption errors improves significantly on WideResNet and ResNext, by $\sim 4\%$ and $\sim 3\%$, respectively.

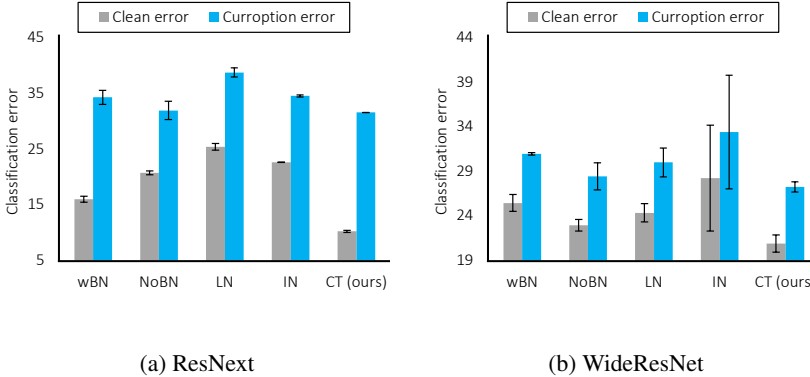

(a) ResNext                    (b) WideResNet

Figure 3: CIFAR-10-C error. For both the models, CT achieves lower corruption errors.

### 4.3 REGULARIZING A BATCH NORMALIZED MODEL USING CT LEADS TO A HIGHER ROBUSTNESS

Here we compare our counterbalance teacher model with recent approaches on robustness, multi-domain generalization, and corrupted point cloud classification using different data sets and models.

**Robustness to common data corruptions.** We compare the robustness of our CT model to two main groups of the recent approaches on robustness: 1) covariate shift adaptation and 2) data augmentation-based methods on CIFAR-10-C and CIFAR-100-C (Hendrycks and Dietterich, 2019). Covariate shift adaptation approaches (Benz et al., 2021; Schneider et al., 2020) are designed with the assumption that there exist some samples from (or close to test distribution) which can be used to calibrate the parameters of the batch normalization layers. However, we show that these approaches work only if BN statistics are adapted to a particular corruption. This is unrealistic, especially at inference time, as we often do not have prior knowledge of corruption types that may or may not occur. Additionally, we show adapting batch norm statistics to a single or a few distortion types can often fail to generalize to unseen distortions (Geirhos et al., 2018; Vasiljevic et al., 2016) which is also the case for data augmentation-based methods.

**CT vs. covariate shift adaptation methods for robustness.** We consider three scenarios for adaptation of batch norm statistics; Let $C = \{c_i \mid i \in Z\}$ be the set of corruptions that can be applied to the input of the network (we assume that the corruption severity level (Hendrycks and Dietterich, 2019) is 3 in all scenarios):

(I) **Adapt-one-test-one**: BN statistics are adapted to a batch of samples from each corruption type $c_i$ at a time, and classification error is computed on the same $c_i$. This is repeated for each corruption type and classification error is averaged over each type (Benz et al., 2021; Schneider et al., 2020). However, it is not trivial to differentiate samples based on corruption type and adapt each separately at test time.

(II) **Adapt-one-test-all**: BN statistics are adapted to a batch of samples from a randomly selected $c_i$, and the classification error is averaged over $C$. This experiment reveals the caveats of adaptation on a single corruption, where model fails to generalize to other types of corruptions.

(III) **Adapt-all-test-all**: BN statistics are adapted to a batch of samples from a *random combination* of samples from $C$. The goal is to aid adaptation of statistics by looking at samples from "all" corruption types. The classification error is then averaged over $C$. This scenario is more realistic than (I) and (II), but still requires a batch of samples from the target domain.

In Figure 4, we shed light on a more realistic scenario of batch norm statistics adaptation. We assume access to some extra samples, but not their corruption types (i.e., `adapt-all-test-all`). Surprisingly, we regardless of batch size, such methods perform significantly worse on both clean and corruption errors compared to our CT method on CIFAR-10-C with WideResNet model. Further analysis on how adaptation-based methods cause generalization failures are included in Table 6.

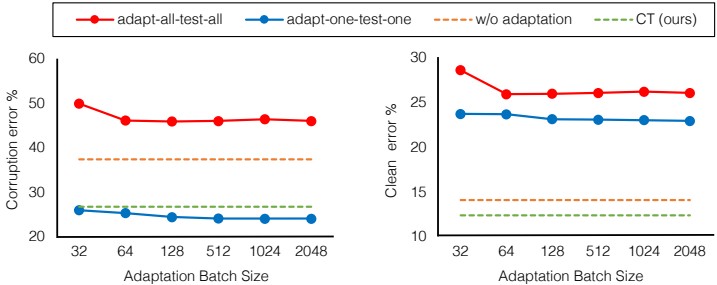

Figure 4: Corruption (left) and clean (right) errors by increasing adaptation batch size on CIFAR-10-C. Covariate shift adaptation-based methods fail in a more realistic setting (`adapt-all-test-all`) i.e. updating the batch norm statistics using randomly selected batches of samples from the target domain, while CT performs significantly better without requiring the target domain knowledge/statistics.

|  |  | SimCLR | BYOL | Barlow Twins | CT (ours) |
|---|---|---|---|---|---|
| CIFAR-10-C | AllConvNet | 30.87 | 34.12 | 34.35 | **14.0±0.55** |
|  | WideResNet | 23.12 | 26.04 | 25.80 | **11.6±0.15** |
| CIFAR-100-C | AllConvNet | 57.51 | 63.76 | 60.07 | **39.8±0.06** |
|  | WideResNet | 57.59 | 56.40 | 55.88 | **33.7±0.12** |

Table 1: CT's mean corruption error (mCE) compared to common self-supervised and contrastive learning methods on CIFAR-10-C and CIFAR-100-C. All SSL models are first pre-trained (self-supervised), then fine-tuned (supervised).

**Comparing the robustness of CT to self-supervised methods on common input corruptions.** In this experiment, we evaluate how recent self-supervised and contrastive learning methods perform on unseen input corruptions: SimCLR (Chen et al., 2020), BYOL (Grill et al., 2020), and Barlow Twins (Zbontar et al., 2021). As reported in Table 1, our method outperforms all self-supervised methods (with fine-tuned encoders) based on the mean corruption error for both CIFAR-10-C and CIFAR-100-C by a large margin. For all the methods in Table 1 we use the same individual augmentations (same operations as in AutoAugment) as in other experiments. See appendix B.2 for linear evaluation results. Concretely, we note that these self-supervised representation learning methods proceed in two stages: pre-training an encoder via self-supervision; and learning a linear classification head on top of the pre-trained encodings. To make it a fair comparison, our modification was to convert this two-stage process into a single joint-training procedure, by using an objective that combines these two terms: (1) a contrastive loss term that uses label information to select positive and negative examples for training the encoder (similar to Supervised Contrastive Learning by (Khosla et al., 2020)); and (2) a cross-entropy loss term for the downstream classification task. In this way, both the encoder and classifier benefit from labeled supervision.

**Comparing CT to data augmentation-based methods.** Hendrycks et al. (2019b) has studied the effectiveness of different data augmentation techniques on making a model robust to common data corruptions. They proposed to create a weighted mixture of input samples using multiple different augmentation types (AugMix) and showed this method can outperform other augmentation-based methods on robustness to common corruptions. Since our CT model is inherently data augmentation-independent, *without any augmentation*, it outperforms (by achieving 25.7% mCE while the original mCE is 39.6%) several recent and widely used augmentation-based approaches such as Standard (Hendrycks et al., 2019b), Cutout (DeVries and Taylor, 2017), CutMix (Yun et al., 2019), and Adversarial Training (Madry et al., 2017) on CIFAR-10-C with WideResNet. When CT is used with a few simple data augmentations (AutoAugment (Cubuk et al., 2018) individual operations, but randomly), as shown in Table 2, it outperforms most of the data augmentation-based methods including the recent and advanced ones such as Mixup (Zhang et al., 2017) and AutoAugment (Cubuk et al., 2018), while it achieves comparable results to AugMix without the Jensen-Shannon divergence (JSD) loss on CIFAR-10-C and CIFAR-100-C datasets with different networks.

|  |  | Standard | Cutout | Mixup | CutMix | AutoAug | AdvT | AugMix | CT (ours) |
|---|---|---|---|---|---|---|---|---|---|
| CIFAR-10-C | AllConvNet | 30.8 | 32.9 | 24.6 | 31.3 | 29.2 | 28.1 | 15.0 | **14.0±0.55** |
|  | WideResNet | 26.9 | 26.8 | 22.3 | 27.1 | 23.9 | 26.2 | 11.2 | 11.6±0.15 |
| CIFAR-100-C | AllConvNet | 56.4 | 56.8 | 53.4 | 56.0 | 55.1 | 56.0 | 42.7 | **39.8±0.06** |
|  | WideResNet | 53.3 | 53.5 | 50.4 | 52.9 | 49.6 | 55.1 | 35.9 | **33.7±0.12** |

Table 2: CIFAR-10-C and CIFAR-100-C mean corruption error (mCE) compared to common data augmentation techniques. AdvT, and AutoAug refer to adversarial training, Auto Augment, respectively. Our CT approach outperforms six out of seven methods while it achieves comparable results to AugMix which leverages complex augmentations.

**CT's performance in domain generalization.** In this experiment, we evaluate CT on domain generalization using the VLCS benchmark (Torralba and Efros, 2011). VLCS consists of images from five object categories shared by the PASCAL VOC 2007, LabelMe, Caltech, and Sun datasets, which are considered to be four separate domains. We follow the standard evaluation strategy used in (Carlucci et al., 2019), where we partition each domain into a train (70%) and test set (30%) by random selection from the overall dataset. We use ResNet-18 as the backbone to make a fair comparison with the state-of-the-art. As summarized in Table 3, CT outperforms the state-of-the-art on 3 out of 4 domains and by 1.83% on average.

| Method | Caltech | LabelMe | Pascal | Sun | Average |
|---|---|---|---|---|---|
| DeepC (Li et al., 2018b) | 87.47 | 62.06 | 64.93 | 61.51 | 68.89 |
| CIDDG (Li et al., 2018b) | 88.83 | 63.06 | 64.38 | 62.10 | 69.59 |
| CCSA (Motiian et al., 2017) | 92.30 | 62.10 | 67.10 | 59.10 | 70.15 |
| SLRC (Ding and Fu, 2017) | 92.76 | 62.34 | 65.25 | 63.54 | 70.15 |
| TF (Li et al., 2017) | 93.63 | 63.49 | 69.99 | 61.32 | 72.11 |
| MMD-AAE (Li et al., 2018a) | 94.40 | 62.60 | 67.70 | 64.40 | 72.28 |
| D-SAM (D'Innocente and Caputo, 2018) | 91.75 | 57.95 | 58.59 | 60.84 | 67.03 |
| Shape Bias (Asadi et al., 2019) | 98.11 | 63.61 | **74.33** | 67.11 | 75.79 |
| CT (ours) | **99.21** | **65.87** | 74.10 | **71.20** | **77.60** |

Table 3: Multi-source domain generalization accuracy (%) on the VLCS dataset with ResNet-18 as the base network for classification. All reported numbers are averaged over three runs.

**CT's classification performance on corrupted 3D point cloud data.** In this experiment, we use the RobustPointSet dataset Taghanaki et al. (2020) which is created for analysis of point classifiers in terms of robustness to 3D corruptions. We follow the same training-domain validation setting as in Taghanaki et al. (2020). We train each model without data augmentation on the clean training set and select the best performing checkpoint on the clean validation set for each method. We then test the models on the six distorted unseen test sets. This experiment shows the vulnerability of the models trained on original data to unseen input transformations. As shown in Table 4, PointNet-CT significantly improves classifcation accuracy on most of the shifted test sets, and 1.87% on average compared to original PointNet.

| Method | Original | Noise | Translation | MissingPart | Sparse | Rotation | Occlusion | Avg. |
|---|---|---|---|---|---|---|---|---|
| PointNet | 89.06 | 74.72 | 79.66 | 81.52 | 60.53 | **8.83** | **39.47** | 61.97 |
| PointNet-NoNorm | 87.99 | 78.65 | 79.02 | 76.82 | 72.61 | 6.89 | 35.13 | 62.44 |
| PointNet-CT (ours) | **89.83** | **78.12** | **82.25** | **82.58** | **68.40** | 7.62 | 38.10 | **63.84** |

Table 4: Robust classification accuracy on RobustPointSet. The `Noise` column for example shows the result of training on the `Original` train set and testing with the `Noise` test set. When we train PointNet using our CT method its performance significantly improves on average and particularly on `Noise`, `Translation`, `MissingPart`, and `Sparse` test sets.

## 5 RELATED WORK

**Reducing classification error on common input corruptions.** Hendrycks et al. (2019b) proposed a data augmentation technique which mixes multiple augmented samples using random weights and showed improvements on robust accuracy. Rusak et al. (2020) proposed to mix Guassian noise with adversarial samples for robustness to common distortions. However, as in the real world, the space of distortions and their mixture is not finite, it is not trivial to devise a set of augmentation types that will make a model robust to all distortions. Moreover, making a model robust to certain corruptions via augmentation does not generalize to others. Dodge and Karam (2017) leverage an ensemble approach for robustness against distortions, but they also assume corruptions are known beforehand.

**Batch normalization's vulnerability.** Several works have studied the effect of batch normalization in the context of robustness to adversarial perturbations. However, as the most relevant works to ours, Schneider et al. (2020) and Benz et al. (2021) suggested improving robustness to common data corruptions by updating the batch normalization parameters using statistics calculated form extra test distribution samples. However, extra test samples are not always available, and similar to data augmentation-based methods, these approaches work only when a model is adapted to a specific distortion and tested on the same distortion type.

**Teacher-student models in the context of robustness.** The focus of the few existing methods on leveraging teacher-student models for robustness has been on adversarial perturbations for developing adversarially robust models either by training teacher encoders with adversarial (Goldblum et al., 2020; Papernot et al., 2016) or augmented (Arani et al., 2021) examples. In contrast, our CT method is augmentation-independent. Adversarially trained models might not necessarily be robust to common data corruptions, as we showed in Table 2, our CT method performs significantly better (at least by 12%) on both CIFAR-10-C and CIFAR-100-C datasets across different models compared to the adversarially trained model (AdvT). Even if the augmented samples are crafted by common data augmentations rather than adding adversarial noise, the robustness obtained by the augmentations does not generalize to unseen corruptions, even when they are from the same family of the augmentations a model has seen (Vasiljevic et al., 2016) during training.

**Similarities to self-supervised methods.** Our CT method is similar to self-supervised models (SSL) (Chen et al., 2020; Grill et al., 2020; He et al., 2020) in the sense that SSL also leverages a copy of the same model but for encoding a different view of the input. In SSL, the copy of the model is the same as the main model, while we discard batch normalization layers in our CT model from one of the copies. Hendrycks et al. (2019a) showed that SSL can improve error on distorted data, however their method—and in general existing self-supervised methods—are highly dependent on data-augmentation (inductive bias) for creating different views while our approach does not require extra views of the data. Note that our method does not employ any self-supervised loss function.

## 6 CONCLUSION

In this work, we investigated the effect of BN on model robustness. We provided a theoretical justification in the overparameterized linear regime for how BN compromises OOD generalization performance by encouraging the model to rely on highly predictive, low-variance input features. Then, we proposed a robust representation learning framework – Counterbalancing Teacher (CT) – which leverages a frozen copy of the same model without BN as a teacher to regularize the features learned by a student network to improve generalization. Empirically, we demonstrated that our method's learned representations are robust to common corruptions on a suite of domain generalization tasks.

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

## APPENDIX

### A    ADDITIONAL EXPERIMENTAL DETAILS

We implement our CT variants using Tensorflow Abadi et al. (2016).

#### A.1    MNIST MISSING FEATURES EXPERIMENT

We trained the MNIST models (architecture below) with SGD optimizer (with Tensorflow's default parameters) and batch size of 128 for 100 epochs. For the NoBN model, we simply removed the BN layers and trained all models without any data augmentation.

Table 5: MNIST small CNN architecture with batch normalization layers. Batch normalization layers (in gray) are omitted in the Teacher network.

| #  | Layer |
|----|-------|
| 1  | Conv2D (in=d, out=16, stride=1) |
| 2  | BatchNorm |
| 3  | ReLU |
| 4  | Conv2D (in=16, out=16, stride=1) |
| 5  | BatchNorm |
| 6  | ReLU |
| 7  | MaxPool2D (stride=2) |
| 8  | Conv2D (in=16, out=32, stride=1) |
| 9  | BatchNorm |
| 10 | ReLU |
| 11 | Conv2D (in=32, out=32, stride=1) |
| 12 | BatchNorm |
| 13 | ReLU |
| 14 | MaxPool2D (stride=2) |
| 15 | Dense (nodes=256) |
| 16 | BatchNorm |
| 17 | ReLU |
| 18 | Dense (nodes=c) |
| 19 | Softmax |

#### A.2    COMPARING CT TO OTHER METHODS ON COMMON INPUT CORRUPTION

For both CIFAR-10-C and CIFAR-100-C we train $\mathcal{G}_\theta$ using Adam optimizer with learning rate of 0.0001 and batch size of 64. In the second step, we train $\mathcal{F}_\zeta$ using SGD with Nesterov, initial learning rate of 0.1 and decaying to 0.00001 using cosine scheduler for 300 epochs.

#### A.3    APPLYING CT TO POINT CLOUDS

We trained both $\mathcal{G}_\theta$ and $\mathcal{F}_\zeta$ with Adam optimizer and learning rate of 0.001 (divided by 10 at epochs 50 and 75) and batch size of 32 for 500 epochs.

#### A.4    APPLYING CT TO DOMAIN GENERALIZATION

We trained CT with SGD optimizer with learning rate of 0.001 and momentum of 0.9. We set batch size to 32 and image size to 224x224, and train for 500 epochs.

## A.5 SELF-SUPERVISED BASELINES

We train the self-supervised methods with batch size 256 for 1000 epochs while maintaining the original hyper-parameters. We fine-tune all self-supervised methods for 350 epochs with Adam Kingma and Ba (2014) and with starting learning rate of 0.001 with a cosine decay, and $l_2$ coefficient set to 0.0005 to avoid overfitting.

## B ADDITIONAL EXPERIMENTAL RESULTS

### B.1 BATCH-NORM ADAPTATION ADDITIONAL RESULTS

Batch norm adaptation-based results for robustness to common input corruptions on CIFAR-10-C.

Table 6: Changes in clean error (CLN-E), mean corruption error (mCE), and corruption error with respect to changes in adaptation batch size (BS). In this table, the model's batch normalization layers are only adapted to one corruption (CRP) based on the method presented in Benz et al. (2021). This table shows an example of adapt-one-test-all scenario. All numbers are reported with the WideResNet40-2 model. Adaptation of batch norm statistics to Impulse noise, when observed test-time data is adequate, will improve the robustness to similar corruption types (Gaussian, Shot and Impulse noise). However it significantly reduces generalization to other corruption types. Top row shows original model performance without adaptation. Second row shows `adapt-one-test-one` results.

| CRP | BS | CLN-E | mCE | Gauss. | Shot | Impulse | Defocus | Glass | Motion | Snow | Compress |
|---|---|---|---|---|---|---|---|---|---|---|---|
| None | - | 14.2 | 37.54 | 56.26 | 47.93 | 49.28 | 29.54 | 56.44 | 38.51 | 32.35 | 32.71 |
| a-o-t-o | 32 | 23.83 | 26.07 | 37.72 | 35.03 | 34.82 | 16.84 | 40.03 | 24.86 | 27.86 | 31.1 |
| Impulse | 32 | 29.04 | 49.88 | 45.92 | 41 | 39.52 | 56.97 | 60.46 | 65.79 | 40.31 | 45.51 |
| Impulse | 16 | 33.06 | 55.02 | 51.39 | 45.73 | 43.62 | 62.27 | 66.23 | 71.03 | 45.62 | 51.12 |
| Impulse | 2 | 43.7 | 61.47 | 62.31 | 58.16 | 58.33 | 62.68 | 73.32 | 68.68 | 56.95 | 59.56 |
| Defocus | 32 | 20.39 | 41.98 | 70.47 | 62.49 | 59.07 | 21.54 | 65.94 | 35.58 | 41.16 | 38.9 |
| Defocus | 16 | 21.11 | 44.43 | 72.24 | 64.88 | 58.77 | 24.53 | 69.55 | 38.36 | 43 | 41.64 |
| Defocus | 2 | 44.47 | 62.1 | 74.39 | 70.69 | 68.91 | 51.03 | 74.98 | 54.16 | 64.18 | 68.92 |

### B.2 LINEAR EVALUATION OF SELF-SUPERVISED MODELS

Self-supervised linear evaluation results where encoders are frozen and only classification layer is trained.

Table 7: CT's mean corruption error (mCE) compared to common self-supervised and contrastive learning methods on CIFAR-10-C and CIFAR-100-C. Encoders are frozen and only classification layer is trained (linear evaluation).

| | | SimCLR | BYOL | Barlow Twins | CT (ours) |
|---|---|---|---|---|---|
| CIFAR-10-C | AllConvNet | 58.84 | 65.47 | 58.2 | 16.9 |
| | WideResNet | 58.18 | 53.32 | 59.75 | 13.9 |
| CIFAR-100-C | AllConvNet | 82.91 | 91.2 | 86.16 | 42.6 |
| | WideResNet | 84.08 | 83.38 | 85.61 | 43.1 |

### B.3 PREDICTIVE PERFORMANCE OF MNIST DATASET VARIABLES.

The following analysis is performed to determine whether the dark and light pixels in the MNIST experiment are discriminative. We plot histograms of all pixel values for each class separately. The histograms look very similar, as shown in Figure 5, indicating that the dark or light pixels are not predictive alone. Therefore we add red or blue squares/variables which are both low variance and predictive.

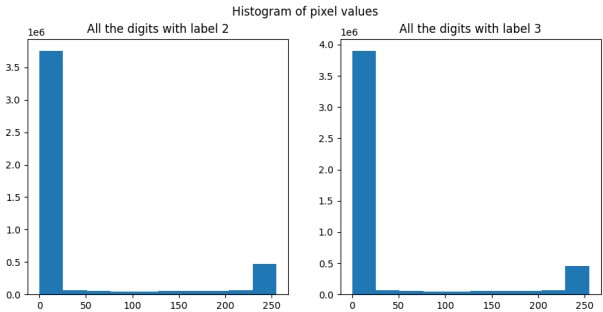

Figure 5: Histogram plots of class '2' vs. '3'.

## C  ADDITIONAL ANALYSIS ON THE EFFECT OF NORMALIZATION IN OVERPARAMETERIZED REGIME

### C.1  SUBTRACTING MEAN DOES NOT CHANGE THE ESTIMATED PARAMETERS

In Section 2.1, we show that dividing each feature by its variance incentivize the model to rely more on features with low variance. Here we show that subtracting mean from each feature and target does not change the estimated parameter in overparametrized regime. Recall that $X \in \mathbb{R}^{n \times d}$ denote the training examples and $Y \in \mathbb{R}^n$ denote their target. Let $\mu_X \in \mathbb{R}^d$ be the mean of features in training data, and $\mu_Y \in \mathbb{R}$ denote the mean of the their targets.

Since $\mu_X$ is a linear combination of the rows of $X$ and row operation does not change the projection matrix; therefore, the following two linear programs have the same solution:

$$\hat{\zeta} = \arg\min \|\zeta\| \qquad\qquad\qquad \hat{\theta} = \arg\min \|\theta\|$$
$$s.t. \quad X\zeta = Y \qquad\qquad\qquad s.t. \quad (X - \mu_X)\theta = Y - \vec{1}\mu_Y$$

### C.2  THE EFFECT OF NORMALIZATION ON MAX-MARGIN CLASSIFIERS

In Section 2.1, we analyze the effect of normalization in linear regression. Here we show that the analysis hold for the max-margin classifiers as well. Soudry et al., Soudry et al. (2018), show that without any explicit regularization, gradient descent on logistic loss converges to the L2 maximum margin separator for all linearly separable datasets. In the overparametrized regime data are completely separable, there are also some investigation that as $d$ increases all the data points serve a support vectors Narang (2020).

Without normalization we have:

$$\hat{\zeta} = \arg\min \|\zeta\|_2^2 \qquad s.t. \quad Y \odot X\zeta \geq 1 \tag{8}$$

Recall that $U$ is a diagonal matrix where $U_{ii}$ is the standard deviation of the $i^{\text{th}}$ feature. After normalization we have:

$$\hat{\beta} = \arg\min \|\beta\|_2^2 \qquad s.t. \quad Y \odot XU^{-1}\beta \geq 1 \tag{9}$$

At the test time we predict $\text{sign}(XU^{-1}\hat{\beta})$, substituting $U^{-1}\beta$ with $\theta$ we have:

$$\hat{\theta} = \arg\min \|U\theta\|_2^2 \qquad s.t. \quad Y \odot X\theta \geq 1 \tag{10}$$

Similar to linear regression scenario normalizing data change the inductive bias of estimator to choose a model that rely more on low variance features. This data-dependent norm lead to good performance in-domain where such features exhibit low variation but will perform poorly out of domain data are corrupted independently or are sampled from a different distribution with different statistic than the training distribution.

