# OpenReview forum: "Counterbalancing Teacher: Regularizing Batch Normalized Models for Robustness"
_ICLR.cc/2022/Conference — ICLR 2022 Submitted_

### Official Review · Reviewer_SoaW · 2021-10-31

**Correctness:** 3
**Technical Novelty And Significance:** 3
**Empirical Novelty And Significance:** 3
**Recommendation:** 6
**Confidence:** 2

**Main Review:**

Strengths:
1. The paper is well-written and well-motivated. The proposed approach is simple, effective and does not require access to the out-of-domain distribution.
2. The proposed approach is empirically evaluated against a number of baselines, including covariate shift adaptation methods, self-supervised methods, and data augmentation-based methods.

Weakness:
1. The theoretical analysis focuses on the limitation of batchnorm, but it is not clear how distillation resolves this problem. If a new model is distilled, one can argue that the effects of batchnorm, along with its limitations, are transferred to the new model, although it does not include the BN layer explicitly—analysis and insights are missing in the proposed counterbalancing teacher in how it address BN's limitation exactly.
2. The authors should clarify why relying on low-frequency features are undesirable. This is an implicit theory in this paper without sufficient justification.

**Summary Of The Paper:**

The paper aims to address the problem of out-of-domain generalization with batch normalization layers. It first identifies the reliance on low-variance features in batch normalization, and proposes a counterbalancing teacher approach to distill from a BN-model to a non-BN model. Experiments verify the hypothesis of the paper and show that the proposed approach outperforms the various baselines on robustness to data corruption and domain generalization. The proposed approach is also evaluated on a 3D point cloud dataset.

**Summary Of The Review:**

I think the simplicity and effectiveness of the proposed approach out-weights the limitations. This approach has potential for further improvements because it does not rely on access to the target data distribution.

---

> ### Author Response · Authors · 2021-11-18
> **Response to Reviewer SoaW**
>
> **Why does distillation help?** As shown in the paper, the model without BN (teacher) learns more robust representations, whereas the model with BN (student) achieves higher clean accuracy. Distillation is meant to achieve the best of both worlds. We use a controllable hyperparameter $\lambda$ to adjust the effect of BN (transfer the behavior of the robust teacher model to the student) on the final model. The effect of $\lambda$ can be seen in our ablation study in response to Reviewer HouJ -- as we increase $\lambda$, or decrease the effect of BN, we achieve higher accuracy on the “None” test set.
>
> **Why not rely on low-frequency features?** It has been shown in prior work (e.g. Geirhos et al. 2018; 2020) that a model’s overreliance on low-variance/low-frequency features (that are often domain- or dataset-dependent) leads to poor generalization performance on new test environments. While this is not necessarily a bad thing, as this behavior improves in-domain performance, our hope is to incentivize the model to focus on “higher-level” features as well to facilitate generalization.

---

> > ### Comment · Reviewer_SoaW · 2021-11-19
> > **Why does distillation help?**
> >
> > > "Distillation is meant to achieve the best of both worlds."
> >
> > I'm not convinced why distillation helps. Could you elaborate how exactly does distillation "achieve the best of both worlds"? If a new model is distilled, one can argue that the effects of batchnorm, along with its limitations, are transferred to the new model, although it does not include the BN layer explicitly

---

> > > ### Author Response · Authors · 2021-11-19
> > > **more clarification**
> > >
> > > - We'd like to start by emphasizing that the model without BN acts as a regularizer, so *the transfer direction is from the model without BN (teacher) to the model with BN (student)*, not the other way around. The model with BN (which has been regularized during training) is used for inference.
> > >
> > >
> > > - The features learned by the BN model are not necessarily bad because they help in improving clean accuracy. The regularization causes the features of the model with BN, which are already good for clean accuracy, to be *also* robust. The $\lambda$ controls the amount of change/regularization towards robustness. Please see our response to Reviewer HouJ for ablation of changing the $\lambda$.

---

### Official Review · Reviewer_yprC · 2021-11-03

**Correctness:** 4
**Technical Novelty And Significance:** 3
**Empirical Novelty And Significance:** 3
**Recommendation:** 8
**Confidence:** 4

**Main Review:**

The paper theoretically analyzes Batch Normalization on linear models, showing that it introduces and inductive bias that makes the model rely more on low-variance features. They argue this high reliance on low-variance features can cause the model to become brittle to distributon shifts where such features are disrupted, while non-normalized models tend to rely to all the input features in a more balanced way, which makes them more robust at the cost of a lower in-distribution accuracy.

The paper proposes a training method where a teacher model is first trained without BN, then a student model with the same architecture but with BN layers is trained both on the task loss and on a regularization loss that pushes its last hidden representation towards the one of the frozen teacher model. They show that this approach preserves in-distribution accuracy of BN while achieving better out-of-distribution accuracy on various tasks.

Question for the authors: are the teacher and the student initialized with the same weights?

Overall the paper is clearly written, the arguments are sound and the empirical evidence is solid.

**Summary Of The Paper:**

The paper argues that Batch Normalization and similar normalization layers can make neural networks more fragile to data distribution shifts and proposes a distillation-based regularization method to ameliorate this issue while maintaining in-domain accuracy.

**Summary Of The Review:**


Overall the paper is clearly written, the arguments are sound and the empirical evidence is solid.

---

> ### Author Response · Authors · 2021-11-18
> **Response to Reviewer yprC**
>
> We thank the reviewer for the positive reception of our work! We do not initialize the student and teacher networks with the same weights -- we will make this point clear in the final version.

---

### Official Review · Reviewer_63kd · 2021-11-04

**Correctness:** 2
**Technical Novelty And Significance:** 3
**Empirical Novelty And Significance:** 2
**Recommendation:** 3
**Confidence:** 4

**Main Review:**

The paper is mostly well written with a clear story, and nicely motivated by a few toy examples. The results outlined in the paper look very promising.

However, especially after the strong focus on batch norm adaptation techniques in the introduction, I expected to see full results and improvements on ImageNet-C, -R, (-A) as common in the referenced papers. In contrast, the experimental results are quite sparse, and comparision to the most relevant competitors (techniques that leverage augmentations during training time) are limited to Table 2 (CIFAR scale). In-depth analysis of the properties of the method, along with ablation studies, is also missing.

In the following I will discuss the most important additional limitations/weaknesses of the paper.

* Methods: The method section in 3.2 lacks clarity. Especially for batch norm, there are a lot of caveats in the exact implementation. Could you please add a precise description how the statistics are estimated, how the student / teacher update works, how student -> teacher update works, etc.? Given the simplicity of the method (which is a huge plus), it should be possible to re-implement the technique based on the information in §3, and I feel like this is not the case currently.
* Motivating examples (§4.1): Especially on such small scale experiments, I find it important to strengthen results by running multiple seeds, and reporting error bars. Could these be added to all experiments in Figure 3, both for the barchart and the line plot? In addition, it could be interesting to add results for an “on the fly” normalization layer such as GroupNorm or LayerNorm to the set of models in Figure 3 (optional). Apart from this, I really like this experiment and how it is used to set the stage for the method.
* Figure 4: The lines should not be connected, there is no obvious relationship between the entries on the x-axis. I suggest a simple scatterplot (one point per method) along with error bars is more suitable in this case.
* Experimental results from §4.3 onwards: It is unclear to me which experiments were re-implemented, and which results are cited from previous work. Could you please let me know for tables 1, 2, 3 and 4 whether you ran all baseline implementations yourself, or whether any of the numbers in the tables is cited? From the appendix, it seems the contrastive learning numbers are reproduced, while the domain generalization results are cited? If that is the case, did you verify any of the baseline results to make sure the experiment setups match?
* Self-supervised learning (Table 1): How do you justify this experiment? If all the labels are known, and the models are fine-tuned, why is it interesting to use self-supervised learning for obtaining the embeddings in the first place? Generally, I think this is a minor result, and should be less prominently mentioned in the paper.
* Instead, It would strengthen the paper to extend Table 2, and add more methods. According to Robust Bench, the [state-of-the art in CIFAR10-C](https://robustbench.github.io/#div_cifar10_corruptions_heading) right now are much better than the reported 17.4%. On a WideResNet-40-2 model, Augmix implemented in the benchmark achives ~11-12%, while 13.5% are reported in the paper. How do you explain the difference? The same applies to CIFAR-C, where Augmix is reported with 35.89%. It would also be good to cite [Diffenderfer](https://arxiv.org/abs/2106.09129) who are state of the art now, although their method is clearly concurrent work (just accepted to NeurIPS'21).
* VLCS: [Gulrajani and Lopez-Paz](https://arxiv.org/abs/2007.01434) showed that on domain generalization benchmarks like VLCS, after sufficient hyperparameter tuning, and by using more suitable and larger models (like a ResNet50), ERM is a very strong baseline. Results in Table 3 show exactly the issue described by Gulrajani and Lopez-Paz. A much stronger result would be a full evaluation on the DomainBed benchmark, if the authors would like to do a strong claim about Domain Generalization. (Also note that all of the cited methods were published before Gulrajani and Lopez-Paz).
* Minor Point: Positioning within the related work: It is known (and already shown on ImageNet-C scale, cf. your references regarding BatchNorm adaptation) that adapting to “shuffled” corruptions (your “adapt-all-test-all” setup) does not work with batch norm adaptation, so this is not a new observation. This should be made clear in §4.2.

A few questions:
* Table 2 suggests that larger models (?) like the WideResNet narrow or even revert the effect of your technique. When moving to full scale data, models get even larger (ResNet50, ResNext101, etc.) and even better augmentation techniques than Augmix (SOTA right now is DeepAugment + Augmix augmentation combined) are used. Could you comment on the limitations of your method and your expectations? For instance, batch norm adaptation fails to work as models and datasets get larger--do you expect that a similar limitation will apply to your technique, that is not visible from the presented results?

**Summary Of The Paper:**

The authors propose a new consistency loss for improving model robustness to common image corruptions. They use a student-teacher training setup where only the student network uses batch normalization at training time. Improvements are shown on small scale corruption datasets (CIFAR-C), a single domain generalization dataset (VLCS), and RobustPointSet.

**Summary Of The Review:**

The technique is simple, well-described, and seems to be effective on the considered toy datasets. What I am missing right now is compelling evidence that it actually works in practice, and in practically relevant settings. For this, several key results are missing:

1. Full scale ImageNet-C (and -R) evaluation with relevant larger models (ResNet50, ResNext101 etc, following prior work). This will make the paper much more interesting for a wider audience interested in corruption robustness. In addition, I suggest runnign on ObjectNet --- BatchNorm adaptation does not work on this dataset, and it would be interesting to see whether CT is effective at mitigating this issue.
2. Comparision to more recent domain generalization techniques, and especially reporting results on more datasets beyond VLCS. To make a claim that the method improves domain generalization, the authors should consider the DomainBed evaluation setting, or a similar, more comprehensive experiment protocol that avoids the issues discussed by Gulrajani & Lopez-Paz (2020). Alternatively, the authors might drop this claim, and move the table as a proof-of-concept to the supplement.
3. I am missing a statement how hyperparameters were tuned. The supplement simply states a few values, but not how the authors arrived at them. In particular, they should be picked according to the *validation error on the training domain*. Ablations over a few crucial parameters will help here (for example, the learning rate). It is unclear why these were not added, given that the results are obtained on fairly small scale datasets.

I think it is feasible and reasonable to expect these kinds of results in an ICLR paper. Point (1) only requires to train models on ImageNet (3 seeds, ideally) *once*, getting scores for the requested setups is easily doable afterwards.  I find it possible that on full ImageNet-C, more recent techniques like DeepAugment+Augmix on larger models (ResNet50) will outperform the proposed technique. Even then, I agree with the authors comment (cf. Table 2) that a huge advantage of the method is the removed need for well-designed data augmentations. However, I feel that this might come at the expense of only being effective in settings where batch norm adaptation is working. Since the technique is effective on corruptions, a great test would be the ObjectNet dataset, where batch norm adaptation fails to work.

Regarding (2), this is not a strong requirement, the authors could equally decide to drop the claim regarding domain generalization.

Point (3) is again crucial---especially Table 2 indicates that the baselines in the paper were not sufficiently tuned.

If the authors manage to add these results (minimally 1, comment on 2, and clarifications on 3, along with addressing the detailed comment in my review), and show a realistic comparisions of tradeoffs to state of the art augmentation based models, I will increase my score. Right now, in my opinion the paper does not meet the bar for ICLR, as the number of truly relevant datasets, and comparisions to the state of the art is limited.

I am happy to address questions/clarify the exact scope of points 1-3 above if this is needed.

---

> ### Author Response · Authors · 2021-11-18
> **Response to Reviewer 63kd [Part 2]**
>
> **Hyper-parameter tuning.**  When there exist published baselines such as the AugMix paper on CIFAR-10-C, CIFAR-100-C, and ImageNet-C, we have followed their setup with small variations, however, we eventually found their setup worked best. When there is no prior published work such as the MNIST experiment, we have tried a reasonable range of different values/choices for the batch size {32, 64, 128, 256}, optimizer {Adam, SGD, Adadelta}, and learning rate {0.0001, 0.001, 0.01, 0.1}. We confirm that we have always chosen the best model based on ​​validation errors on the training domain.
>
> **Regarding the comment** *“on full ImageNet-C, more recent techniques like DeepAugment+Augmix on larger models (ResNet50) will outperform the proposed technique. Even then, I agree with the authors comment (cf. Table 2) that a huge advantage of the method is the removed need for well-designed data augmentations. However, I feel that this might come at the expense of only being effective in settings where batch norm adaptation is working.”*
>
> As mentioned by the reviewer, we would like to emphasize that our method does not require complex and well-designed data augmentation or access to test data, as traditional batch norm adaptation techniques do. Data augmentation can be ineffective because 1) the space of possible augmentations at test time is not known in advance (it could be a single or a mixture of multiple unknown transformations); and 2) even if the test time noise/transformation is known to some extent, a trained model with a certain type of noise (as augmentation) may still be vulnerable to an unseen variant from the same noise family, e.g., as shown by Vasiljevic et al., 2016, training a model by blur augmentations created using certain defocus kernel size degrades performance on sharp images and all other blur types. Batch norm adaptation techniques, on the other hand, only work (as demonstrated in our paper) if we have access to test data, which is not a requirement for our method.

---

> > ### Comment · Reviewer_63kd · 2021-11-29
> > **Re: Response**
> >
> > > As mentioned by the reviewer, we would like to emphasize that our method does not require complex and well-designed data augmentation or access to test data, as traditional batch norm adaptation techniques do [...]
> >
> > Please see my [comment above](https://openreview.net/forum?id=sTkY-RVYBz&noteId=5MgTiD9OxQW) where I listed three simple baselines from the literature that do not require access to the test data or additional data augmentations. These techniques seem to outperform your algorithm on ImageNet-C, and it might be relevant to include them also on your smaller scale experiments.

---

> ### Author Response · Authors · 2021-11-18
> **Response to Reviewer 63kd [Part 1]**
>
> **Results on ImageNet-C and ablation of the loss function.** We did our best to run ImageNet experiments with proper hyperparameter tuning given the short rebuttal period. We show our preliminary results (without tuning) below, which demonstrate a significant decrease in the error of our method over the baseline (ResNet-50) on ImageNet-C. We will run more ImageNet experiments for the final version. The values inside parentheses show the error for the baseline (ResNet-50). Lower is better.
>
> gaussian_noise: 68.931 (79)
>
> shot_noise: 68.600 (80)
>
> impulse_noise: 70.098 (82)
>
> defocus_blur: 79.830 (82)
>
> glass_blur: 90.549 (90)
>
> motion_blur: 81.472 (84)
>
> zoom_blur: 86.221 (80)
>
> snow: 81.198 (86)
>
> frost: 81.431 (81)
>
> fog: 67.978 (75)
>
> brightness: 61.528 (65)
>
> contrast: 62.008 (79)
>
> elastic_transform: 86.668 (91)
>
> pixelate: 81.625 (77)
>
> Jpeg_compression: 79.982 (80)
>
> **Mean C-Error: 76.5 (80.6)**
>
>
> We also ran ablation experiments of the loss terms as shown under Reviewer HouJ.
>
> =======
>
> **How are the student/teacher networks updated?** We apologize for the confusion and will include more details in the final version.
>
> 1) We first fully train the teacher network (the one with all BN layers removed) i.e., $\mathcal{G}_\theta$.
> .
> 2) We freeze the fully trained teacher model.
>
> 3) We start training the student model (the one with BN layers) i.e.,  $\mathcal{F}_\zeta$ while regularizing its final feature vector $r{\mathcal{F}_\zeta}$ (right before the softmax) using the last feature vector from the frozen teacher model  $r{\mathcal{G}_\theta}$ via the second term in Eq. 7. To get feature vectors  $r{\mathcal{F}_\zeta}$  and $r{\mathcal{G}_\theta}$  we pass the same input to both the student and teacher networks. The teacher network is never updated after step 2, and only the student is updated while it is regularized.
>
>
> **Figure 3, run with multiple seeds.** We have updated the results of Table 2 and Figure 3 (now shown as Figure 2a in the revised PDF) to include multiple runs.
>
> **Reproduced and cited results.** We have only replicated contrastive learning results (Table 1) because there have been no reported results on the tasks/datasets that are relevant to our current work. We have also ensured that our replicated experimental setups match those of the cited methods.
>
> **Self-supervised learning (Table 1).** We will move our supervised contrastive learning (SCL) results in Table 1 to the appendix.
>
> **Adding more methods to Table 2.** We followed the setup in the AugMix paper (in terms of architectures, etc.) to ensure a fair comparison to other methods.
>
> Regarding AugMix results, we included two versions of AugMix in our paper: a) with JSD loss (showed as AugMix which archives 11.2% error on CIFAR-10-C) b) without JSD loss (showed as AugMix* which archives 13.5% error on CIFAR-10-C). We removed AugMix* from the table. As suggested by Reviewer HouJ, we ran our method without L2 normalization which led to our method outperforming AugMix as well.
>
> As for the concurrent work of Diffenderfer et al. 2021, they leverage AugMix and **Gaussian noise** augmentations during training. This makes it difficult to compare their experiments on CIFAR-10-C and CIFAR-100-C with ours because Gaussian noise is explicitly a test time transformation in these datasets (so it shouldn’t be used as data augmentation during training).
>
> We also note that the methods in RobustBench used different architectures (ResNet-50, WideResNet-18-2, ResNet-18, WideResNet-70-16), whereas we tried to be consistent with the AugMix paper (WideReNet-40-2, AllConvNet, and ResNext), which tested many different methods on the same architectures.
>
> **VLCS.** We chose VLCS over others in the DomainBed paper because it appeared that only on VLCS did people use the same train/val/test splits. There are many different results in the literature for other datasets such as HomeOffice and PACS because each paper used its own train/val/test splits. Therefore, we take the reviewer’s point and will move the VLCS experiment to the appendix.
>
> Regarding the VLCS results in the DomanBed paper (Gulrajani & Lopez-Paz (2020)); all of the methods listed in our paper, as well as the majority of methods in the literature (including the ones that the DomainBed paper compares to), are based on ResNet-18 on the VLCS dataset; however, the DomainBed paper uses ResNet-50, which is a bigger model. We attempted to be consistent with the literature in order to demonstrate the efficacy of the proposed method. However, we take the reviewer’s point and will move the VLCS experiment to the appendix.

---

> > ### Comment · Reviewer_63kd · 2021-11-18
> > **Re: Results on ImageNet-C and ablation of the loss function**
> >
> > Thanks for the additional experimental results.
> >
> > You report a baseline mCE of 80.6% on ImageNet, vs. 76.5% after application of your method.
> >
> > This falls short of e.g. the baseline ResNet50 model trained on standard ImageNet (76.7% mCE), a standard Resnet-50 trained with Group Normalization (72.0% mCE) or with Fixup initialization (72.0% mCE), or with Batch Norm adaptation at Batch Size 1 (71.39% mCE), cf. Schneider et al. 2020.
> >
> > I suspect that you did not train the model long enough? In any case, could you report the ImageNet-C results for your baseline model after full batch norm adaptation, to get a sense of the overall improvement that is achievable?

---

> > > ### Author Response · Authors · 2021-11-19
> > > **ImageNet baseline**
> > >
> > > Thank you for the follow up comment. We did not train the baseline, we used the reported mCE from the AugMix paper which is 80.6 for the ResNet-50 baseline.

---

> > > > ### Comment · Reviewer_63kd · 2021-11-19
> > > > **Re: ImageNet baseline**
> > > >
> > > > This is good to know. I assume that you build your experiment on top of the AugMix reference implementation then? Could you share more details (implementation, hyperparameters, code, if you build on publicly available code) on how you arrived at 76.5% mCE, and your interpretation of this number?

---

> > > > > ### Author Response · Authors · 2021-11-19
> > > > > **Re: ImageNet baseline**
> > > > >
> > > > > An anonymized code can be accessed [here](https://anonymous.4open.science/r/CT-152C/). The decrease in mCE is due to learning more robust features and may further improved with a careful tuning. We will add more results to the final version of the paper.

---

> > ### Comment · Reviewer_63kd · 2021-11-29
> > **Re: Method**
> >
> > > How are the student/teacher networks updated? We apologize for the confusion and will include more details in the final version.
> >
> > As far as I see, this did not happen with the updated paper yet? Cf. the [diff](https://openreview.net/revisions/compare?id=sTkY-RVYBz&left=MtXXzs65oEP&right=8T4g9OqSmj&pdf=true).
> >
> > I would encourage you to simplify notation (e.g. by dropping the parameter subscripts $\theta$ and $\zeta$ which are never used as far as I see), and -- reiterating my original review -- make it very clear how the method is implemented. On reading the section again, there are also some issues with introducing notation, e.g. $p^i_G$ is never introduced, to name one example.
> >
> > In case you plan to address this point, could you please paste the updated section here before the comments close?
> >
> > ---
> >
> > Regarding your description above,
> >
> > > We first fully train the teacher network (the one with all BN layers removed)
> >
> > Could you provide details how this would work on ImageNet scale without additional means for careful initialization? Or is this a limitation of the method? How are the weights in the network initialized, and are there potential problems when training the network without batch norm?

---

> > > ### Author Response · Authors · 2021-11-29
> > > **Edited Method Section**
> > >
> > > - The modified method section is pasted below. We kept $\theta$ and $\zeta$ in the method section for consistency because we used them in section 2.1. Please let us know if you want us to make more changes.
> > >
> > > - We did not use any initialization techniques etc. when training the teacher network on ImageNet. We simply used small learning rates, such as 0.0001, as in our previous experiments. We will add this to the implementation details.
> > >
> > > --------
> > >
> > > **************** The modified Method Section starts here ******************
> > >
> > > Our approach consists of two separate steps: 1) fully training and freezing the counterbalancing teacher (CT), 2) training the Student while regularizing it using the frozen CT.
> > >
> > >
> > > ### Training the counterbalancing teacher (CT)
> > >
> > > For an arbitrary neural network encoder $\mathcal{F}$ parametrized by $\zeta$ with one or more batch normalization layers, we create a clone of the network and remove all the batch norm layers from the cloned network. We refer to the cloned network as $\mathcal{G}$ and its parameters as $\theta$. Henceforward we refer to $\mathcal{G}_\theta$ as the Counterbalancing Teacher or simply CT, and $\mathcal{F}_\zeta$ as the Student. In the first step, we train the CT using the classification ($k$ class) objective of the task in hand. In our experiments, we minimize the cross-entropy loss as a supervised classification objective for the CT:
> > >
> > > **[Eq. 5 here]**
> > >
> > > where $p$ refers to predictions of the CT. Once the CT is fully trained, we freeze it to be used for regularizing the Student in the next step.
> > >
> > >
> > >
> > > ### Regularizing the Student for robustness using the counterbalancing teacher (CT)
> > >
> > > In the second step, we start training the Student by minimizing the cross-entropy loss and regularizing it (simultaneously) using the frozen CT. Inspired from the work of~\cite{huang2017arbitrary}, we propose the following objective to regularize the representations of the Student (i.e., $\mathcal{F}_{\zeta}$):
> > >
> > > **[Eq. 6 here]**
> > >
> > > where $r_{\mathcal{F}_{\theta}} \in \mathbb{R}^h$  and
> > >
> > > $r_{\mathcal{G}_{\theta}} \in \mathbb{R}^h$  refer to the  $h$ dimensional representations learned by the Student and  CT, respectively. Therefore, the total loss to train the Student becomes:
> > >
> > >
> > > **[Eq. 7 here]**
> > >
> > > The $\lambda$ controls the amount of regularization.

---

> ### Author Response · Authors · 2021-11-27
> **Please let us know if you have more questions.**
>
> As we are approaching the end of the discussion phase, we would greatly appreciate it if you could let us know if you have any additional questions or any concerns that you want us to further address.

---

> ### Comment · Reviewer_63kd · 2021-11-29
> **Final discussion points**
>
> Dear authors,
>
> thanks for the updated numbers on ImageNet-C. This triggers the following question: You use a baseline model with 80.6% mCE and improve this model to 76.5% mCE, i.e. you get a 4.1% point improvement in mCE using the proposed training technique.
>
> Let's compare this to a few published baselines -- unfortunately I do not know a paper using the suboptimal baseline that gets 80.6% mCE, the standard ResNet50 pytorch checkpoint starts off slightly below, at 76.7% mCE. I looked into the references you cite for BatchNorm adaptation, and [Schneider et al. (2020)](https://arxiv.org/pdf/2006.16971.pdf) report in Table 3 and Table 12 the following reference results:
>
> | Model | mCE | improvement
> |--|--|--|
> | ResNet50 (BN), non-adapted | 76.7% mCE | baseline
> | ResNet50 (GroupNorm) | 72.4% mCE | -4.3% points |
> | ResNet50 w/ Fixup initialization | 72.0% mCE | -4.7% points |
> | ResNet50 (BatchNorm), adaptation w/ Batch Size 1* | 71.4% mCE | -5.3% points |
>
> All of these three published (and arguably quite simple) baselines outperform your reported result in terms of absolute improvement in mCE percentage points. I should stress that *none* of the three techniques above use data augmentation, or assume knowledge about the domain label (this is different from full batch norm adaptation, which gets even better).
>
> My thought is that these are (a) three relevant baselines that were not considered on the smaller scale experiments as far as I saw and (b) that the improvement we can expect when scaling your technique does not outperform these conceptually simpler baselines (note that Batch norm adaptation does not even require to re-train the model).
>
> ---
> *) Table 12 is an ablation table with multiple hyperparameter choices, but see Figure 11 for the optimal pseudo-batchsize of N=16.

---

> > ### Author Response · Authors · 2021-11-29
> > **Re: Final discussion points**
> >
> > > unfortunately I do not know a paper using the suboptimal baseline that gets 80.6% mCE
> >
> > Please see Table 2 (first row) of the [AugMix](https://arxiv.org/pdf/1912.02781.pdf) paper; they reported an 80.6 on ResNet-50 with standard data augmentations. They **excluded** simple data augmentations which overlap with ImagNet-C so they got 80.6 for the baseline (see below for more detail). Our configuration is very similar to theirs.
> >
> > > I should stress that none of the three techniques above use data augmentation
> >
> > Please note that even simple random data augmentations, which are commonly used, can overlap with ImageNet-C test sets and thus improve the mCE. More information can be found on page 4 of the [AugMix](https://arxiv.org/pdf/1912.02781.pdf) paper, but they mentioned *"we exclude operations which overlap with ImageNet-C corruptions. In particular, we remove the **contrast**, **color**, **brightness**, **sharpness**, and Cutout operations so that our set of operations and the ImageNet-C corruptions are disjoint"*.
> >
> > [Schneider et al. (2020)](https://arxiv.org/pdf/2006.16971.pdf), didn't mention what kind of (simple) data augmentations they used to train their baseline, Group Norm, and Fixup methods. It is critical to exclude the augmentations mentioned above when one tests on ImageNet-C.

---

> > > ### Comment · Reviewer_63kd · 2021-11-29
> > > **Re: Re: Final discussion points**
> > >
> > > > Please see Table 2 (first row) of the AugMix paper; they reported an 80.6 on ResNet-50 with standard data augmentations. They excluded simple data augmentations which overlap with ImagNet-C so they got 80.6 for the baseline (see below for more detail). Our configuration is very similar to theirs.
> > >
> > > Thanks, I am aware where this number is coming from, my original statement was misleading. I meant to say: I am not aware of a paper that uses this particular baseline and applies methods like GroupNorm, Fixup init or single sample batch norm adaptation.
> > >
> > > Hence, I would encourage you to either (1) run the respective baselines on  your ResNet50 version or (2) re-run your method using the [PyTorch reference implementation](https://github.com/pytorch/examples/tree/master/imagenet), so that you can cite the respective numbers to compare to that I listed.
> > >
> > > Does this clarify my concern?
> > >
> > > Irrespective of that, I suspect that the difference in the baseline model/implementation you start with will not drastically affect the message that all three simple baselines at least slightly outperform the training method you propose.
> > >
> > > > Schneider et al. (2020), didn't mention what kind of (simple) data augmentations they used to train their baseline, Group Norm, and Fixup methods. It is critical to exclude the augmentations mentioned above when one tests on ImageNet-C.
> > >
> > > This is the case for all numbers I put in my comment, please carefully check the reference again.
> > >
> > > For GroupNorm and Fixup, please see the paragraph "Group Normalization and Fixup Initialization performs better than non-adapted batch norm models, but worse than batch norm with covariate shift adaptation" which references the respective setups used to train the GroupNorm/Fixup models, which perform standard ImageNet training w/o augmentations (besides horizontal flipping of images, which is true for all baselines).
> > >
> > > For the vanilla/baseline model, the respective code is available [in the pytorch github repository](https://github.com/pytorch/examples/blob/151944ecaf9ba2c8288ee550143ae7ffdaa90a80/imagenet/main.py#L205-L215). Can you clarify why you think that any of the results I linked were obtained with additional augmentations?
> > >
> > > The batch norm adaptation method is applied to the **baseline, vanilla** ResNet50 model and hence also does not use any augmentation. This is pretty clearly outlined at multiple stages in the paper. **If** you apply additional augmentations on top, the results improve further, you see additional numbers with e.g. AugMix training in Table 12.

---

### Official Review · Reviewer_HouJ · 2021-11-04

**Correctness:** 3
**Technical Novelty And Significance:** 3
**Empirical Novelty And Significance:** 2
**Recommendation:** 5
**Confidence:** 4

**Main Review:**

**Strengths:**

- I love the clarity of exposition, especially section 2.1 where the insight of data-dependent versus independent norms is presented.  Also the experiments in section 4, which begin with a very clear description of the hypotheses to be tested.

- The authors perform an extensive set of experiments for different models, datasets (and types).  Particularly impressive was the comparison of CT to other data augmentation-based methods

**Weaknesses:**

- Section 4.1 is a bit oddly presented.  In particular, the red vs blue experiment.  The third paragraph states:
  > "Ideally, a classifier should not rely only on the dominant features to
predict the class. However, as shown in the second row of Figure 2, the network trained with batch normalization quickly picks up those low variance clues, which in this case is either the red or the blue
square. However, the same network without normalization takes into account other features as well.".

I find it hard to draw a conclusion from this experiment.  Have the authors really controlled all the sources of variation in this experiment?  Are the synthetic features shown really the low(est) variance features that disambiguate 2 from 3 in MNIST?  I imagine there are many light / dark pixels that are also low-variance & predictive.  A histogram of variances would be helpful here in establishing the validity of the design.

- Figure 4 is oddly presented.  The X-axis is categorical (different normalization for layers of the wide ResNet), and are not ordinal; presenting this as a line plot suggests that they are.  A grouped bar-plot or box-whisker plot (or violin plot) would be more direct.  Another issue is that the results of Figure 4 are discussed in section 4.2  with a condition not depicted:
> As shown in Figure 4, the model with no normalization (NN) outperforms all the three types of normalization when evaluated on corrupted data
No data for (NN) is presented in Figure 4.  Was the name changed?

- The central insight of the paper is that normalization of features leads to an over-reliance on low-variance features.  Yet after presenting the $L_{ct}$ regularizer, the authors state that they $l_2$ normalize both $r_{G_{\theta}}$ and $r_{F_{\zeta}}$.  It seems very strange to me that a corrective measure against normalization must still require normalization.  Is there no other way to account for the different scales of learned features?  *Should* the method account for them?  I'd like to see a version of the CT loss where different losses for equation (6) are used.

- Table 2 was my favourite part of the whole paper, but I was sad to read that the authors only reported point estimates of mCE.

- Given how the student-teacher regularized loss is defined in equation [7], it seems odd that the authors do not present an ablation study where they reduce the regularization effect tuning $\lambda * \mathcal{L}_{ct}$ as $\lambda \rightarrow 0$


**Summary Of The Paper:**

The authors study the effects of BatchNorm layers on model robustness.  They demonstrate that models trained with BatchNorm are likely to rely on low-variance features which are performant in-domain, but are not helpful for out of domain data.  Leveraging this observation, they propose a student-teacher method (Counterbalancing Teacher) that creates a copy of the same model with all BatchNorm layers removed.  The demonstrate that this leads to very competitive performance with other techniques for reducing the performance hit of out domain evaluation: self-supervised methods and data augmentation methods.


**Summary Of The Review:**

The authors present evidence that BatchNorm leads to a preference for parameters that minimize a data dependent norm, and demonstrate a simple student-teacher training method to correct for these effects.  There are a few alterations I'd really like to see, though I still believe that, given the required corrections, there is enough value in this particular insight and the experiments to see it published.  I would have dearly loved to see more emphasis on other measures of uncertainty for the networks (cf. https://github.com/uncertainty-toolbox/uncertainty-toolbox) rather than looking strictly at accuracy.  If my other criticisms can be addressed, I will increase my score.

---

> ### Author Response · Authors · 2021-11-18
> **Response to Reviewer HouJ**
>
> **The variance of other pixels in MNIST and if they are predictive.** The reviewer brought up a good point about examining the variance of the other pixels in the red vs. blue MNIST experiment. We computed the light and dark pixel statistics (threshold: 20 for pixel values in the range [0-255]). The outcomes are shown below. The statistics show that these features are not truly predictive, as the statistics for both classes are (almost) the same.
>
> **Class 2:**
>
> mean and std of light pixels:  186.73 ± 76.72
>
> mean and std of dark pixels:  0.18 ± 1.40
>
> the average number of light pixels:  157
>
> the average number of dark pixels:  626
>
> ============
>
> **Class 3:**
>
> mean and std of light pixels:  184.16 ± 77.40
>
> mean and std of dark pixels:  0.18 ± 1.41
>
> the average number of light pixels:  151
>
> the average number of dark pixels:  632
>
> The histogram of all pixel values for class ‘2’ vs ‘3’ is also similar (and thus not predictive), as shown here: https://imgur.com/a/TZxhdW9. We will include this analysis in the new version of the paper.
>
> **L2 Normalization.** Previously, we only used L2 normalization to reduce the magnitude of the loss values. The method works well without L2. If we discard L2, we simply need to use a smaller multiplier ($\lambda$) for the CT term in the total loss as the CT term might become very large. Current results are without using L2 normalization.
>
> **Table 2.** We have now added the results of multiple runs to the table.
>
> **Ablation study on $\lambda$.** We conducted an ablation study of the two loss terms (Eq. 7) on the MNIST experiment. As shown below, increasing the \lambda value (the CT term for the teacher's contribution) improves results on the *“None”* test set. We will add these results to the final version of the paper.
>
> $\lambda$ = 0, RB: 100 | R: 56.98 | B: 100 | None: 50.99
>
> $\lambda$ = 20, RB: 100 | R: 96.93 | B: 99.58 | None: 74.11
>
> $\lambda$ = 50, RB: 99.95 | R: 99.94 | B: 88.91 | None: 86.04
>
> $\lambda$ = 70, RB: 100 | R: 99.32 | B: 98.02 | None: 88.91
>
> $\lambda$ = 90, RB: 99.74 | R: 99.58 | B: 99.58 | None: 98.91
>
> **Uncertainty analysis.** We appreciate the reviewer's excellent suggestion to look into other uncertainty measures, and will add  them into the final version.

---

> > ### Author Response · Authors · 2021-11-22
> > **Response to Reviewer HouJ; More analysis on the white/dark pixles**
> >
> > Thank you for clarifying. To ensure that the white/dark pixels are not predictive, we added Gaussian noise (0, 0.2) to the samples *before* adding the red/blue squares and repeated the MNIST experiment (please see the new images [here](https://imgur.com/a/2mUFaOD)). Random noise should change the variance of those dark/light subsets (if any) and make them less predictive. Therefore, the only low variance and predictive features will be the coloured squares. The $\lambda$ ablation results on the "None" test set after adding noise are as follows:
> >
> > ---
> >
> > $\lambda=90: 98.25 \pm 0.87$
> >
> > $\lambda=70: 93.73 \pm 3.90$
> >
> > $\lambda=50: 87.4 \pm 9.0$
> >
> > $\lambda=20: 83.3 \pm 3.34$
> >
> > $\lambda=0: 50.9 \pm 0.33$

---

> ### Author Response · Authors · 2021-11-27
> **Please let us know if you have more questions.**
>
> As we are approaching the end of the discussion phase, we would greatly appreciate it if you could let us know if you have any additional questions or any concerns that you want us to further address.

---

> ### Author Response · Authors · 2021-11-28
> **Response to Reviewer HouJ [Uncertainty analysis results]**
>
> Thank you again for suggesting that we conduct an uncertainty analysis (we leveraged the uncertainty-toolbox). We used the [MC Dropout](https://arxiv.org/pdf/1506.02142.pdf) method to compute the mean standard deviation of predictions across 500 sampling iterations. Please see [this figure](https://imgur.com/a/ncVHyo4) for qualitative results. The quantitative results for our CT method and the baseline on the MNIST dataset are reported below. As can be seen, our CT method achieves both higher classification accuracy and better uncertainty results on the "None" test set. We will add these results to the final version of the paper.
>
>
> **Baseline**
>
> ```
> ===================== Accuracy Metrics =====================
>   MAE           0.486
>   RMSE          0.690
>   MDAE          0.004
>   MARPD         196.451
>   R2           -0.907
>   Correlation   0.707
> =============== Average Calibration Metrics ================
>   Root-mean-squared Calibration Error   0.378
>   Mean-absolute Calibration Error       0.341
>   Miscalibration Area                   0.344
> ========== Adversarial Group Calibration Metrics ===========
>   Mean-absolute Adversarial Group Calibration Error
>      Group Size: 0.11 -- Calibration Error: 0.358
>      Group Size: 0.56 -- Calibration Error: 0.345
>      Group Size: 1.00 -- Calibration Error: 0.341
>   Root-mean-squared Adversarial Group Calibration Error
>      Group Size: 0.11 -- Calibration Error: 0.392
>      Group Size: 0.56 -- Calibration Error: 0.383
>      Group Size: 1.00 -- Calibration Error: 0.378
> ==================== Sharpness Metrics =====================
>   Sharpness   0.014
> =================== Scoring Rule Metrics ===================
>   Negative-log-likelihood   6075.400
>   CRPS                      0.481
>   Check Score               0.241
>   Interval Score            4.953
>
> Classification accuracy (%) on different test sets: RB: 100, R: 50.53, B: 100, None: 50.53
>
> ```
>
>
>
> **CT (ours)**
>
> ```
> ===================== Accuracy Metrics =====================
>   MAE           0.056
>   RMSE          0.145
>   MDAE          0.009
>   MARPD         107.622
>   R2            0.915
>   Correlation   0.962
> =============== Average Calibration Metrics ================
>   Root-mean-squared Calibration Error   0.142
>   Mean-absolute Calibration Error       0.116
>   Miscalibration Area                   0.117
> ========== Adversarial Group Calibration Metrics ===========
>   Mean-absolute Adversarial Group Calibration Error
>      Group Size: 0.11 -- Calibration Error: 0.125
>      Group Size: 0.56 -- Calibration Error: 0.118
>      Group Size: 1.00 -- Calibration Error: 0.116
>   Root-mean-squared Adversarial Group Calibration Error
>      Group Size: 0.11 -- Calibration Error: 0.153
>      Group Size: 0.56 -- Calibration Error: 0.146
>      Group Size: 1.00 -- Calibration Error: 0.142
> ==================== Sharpness Metrics =====================
>   Sharpness   0.064
> =================== Scoring Rule Metrics ===================
>   Negative-log-likelihood   -2.843
>   CRPS                      0.041
>   Check Score               0.021
>   Interval Score            0.238
>
>
> Classification accuracy on different test sets: RB: 99.84, R: 98.59, B: 99.58, None: 96.77
> ```

---

### Author Response · Authors · 2021-11-18
**General response**

We thank the reviewers for their insightful comments and suggestions! We first address common concerns, then address individual reviews separately.

**Odd Presentation of Figure 4.** We have fixed Figure 4 to address the points of confusion (it is now Figure 3 in the revised PDF), changed it to be a bar chart instead of a line plot, and have included error bars for multiple runs.

---

### Decision · Program_Chairs · 2022-01-20

**Decision:**

Reject

**Comment:**

The authors have proposed a new consistency loss for improving model robustness to common corruptions. With a student-teacher training setup, only the student network uses batch normalization at training time. Improvements are shown on small scale corruption datasets (CIFAR-C), a single domain generalization dataset (VLCS), and RobustPointSet.

Though, positive feedback were given on the quality of the story telling, and on an interesting motivation by a few toy examples, some concerns remained among the reviewers.
In particular applicability of the method as model and data sizes increases, e.g., on ImageNet-C, was questioned.
After Additional results were provided by the authors, the method seems to break as scales increases.
The way relevant baselines from previous work was also judged light and should be improved.
Hence, the paper could be improved to include more comparisons and more convincingly showing advantages of the method.